# Expression of modified FcγRI enables myeloid cells to elicit robust tumor-specific cytotoxicity

Leen Farhat-Younis[1], Manho Na[1], Amichai Zarfin[1], Aseel Khateeb[1], Nadine Santana-Magal[1], Alon Richter[1], Amit Gutwillig[1], Diana Rasoulouniriana[1], Annette Gleiberman[1], Lir Beck[2], Tamar Giger[3], Avraham Ashkenazi[4], Adi Barzel[5], Peleg Rider[1], Yaron Carmi[1]*

[1]Department of Pathology, School of Medicine, Tel Aviv University, Tel Aviv, Israel; [2]Department of Human Molecular Genetics and Biochemistry, Tel Aviv University, Tel Aviv, Israel; [3]Department of Molecular Cell Biology, Weizmann Institute, Rehovot, Israel; [4]Department of Cell and Developmental Biology, School of Medicine, Tel Aviv University, Tel Aviv, Israel; [5]Department of Biochemistry Molecular Biology, George S. Wise Faculty of Life Sciences, Tel Aviv University, Tel Aviv, Israel

*For correspondence:
yaron.carmi@gmail.com

**Abstract** Despite the central role of T cells in tumor immunity, attempts to harness their cytotoxic capacity as a therapy have met limited efficacy, partially as a result of the suppressive microenvironment which limits their migration and activation. In contrast, myeloid cells massively infiltrate tumors and are well adapted to survive these harsh conditions. While they are equipped with cell-killing abilities, they often adopt an immunosuppressive phenotype upon migration to tumors. Therefore, the questions of how to modify their activation programming against cancer, and what signaling cascades should be activated in myeloid cells to elicit their cytotoxicity have remained unclear. Here, we found that activation of IgM-induced signaling in murine myeloid cells results in secretion of lytic granules and massive tumor cell death. These findings open venues for designing novel immunotherapy by equipping monocytes with chimeric receptors that target tumor antigens and consequently, signal through IgM receptor. Nonetheless, we found that myeloid cells do not express the antibody-derived portion used to recognize the tumor antigen due to the induction of an ER stress response. To overcome this limitation, we designed chimeric receptors that are based on the high-affinity FcγRI for IgG. Incubation of macrophages expressing these receptors along with tumor-binding IgG induced massive tumor cell killing and secretion of reactive oxygen species and Granzyme B. Overall, this work highlights the challenges involved in genetically reprogramming the signaling in myeloid cells and provides a framework for endowing myeloid cells with antigen-specific cytotoxicity.

## eLife assessment

The findings are **fundamental** for understanding IgM signaling in myeloid cells. The work is **compelling** in its ability to manipulate and harness myeloid cells to further anti-tumor immunity.

## Introduction

Clinical and experimental data have highlighted the important role of the immune cell function and composition in determining tumor progression or eradication (*Coussens et al., 2013*; *Fridman et al., 2012*). While a high prevalence of tumor-infiltrating T cells is associated with improved prognosis and

survival (*Thorsson et al., 2018*; *Bruni et al., 2020*), infiltration of myeloid cells is often associated with poor prognosis and tumor refractory (*Greten and Grivennikov, 2019*; *Hanahan and Coussens, 2012*). Based on these findings, vast scientific effort is executed to increase the host's T cell response to cancer cells (*Fritz and Lenardo, 2019*; *Waldman et al., 2020*). In most cases however, tumor-reactive T cells generated spontaneously by the host bear low to moderate affinity ($10^{-4}$M- $10^{-6}$M) (*Hoffmann and Slansky, 2020*; *Reuben et al., 2020*), which is below the activation threshold needed for therapeutic effect, and attempts to increase TCR affinity using genetical engineering are often limited by induction of cross-reactivity (*Johnson et al., 2009*; *Casucci et al., 2015*).

To overcome these limitations, T cells can be genetically modified to express Chimeric Antigen receptor (CAR) consisting of Single Chain Fragment Variable (scFv) for tumor antigen recognition fused to T cell signaling domains (e.g. CD3 $\zeta$ , CD28) (*Newick et al., 2017*; *Rafiq et al., 2020*). Engineered cells are reinfused into the patient, providing the host's immune system with T cells capable of specifically recognizing tumor antigens in an MHC-independent fashion. Indeed, this strategy benefits from overcoming the complex, multistep, and highly regulated activation process of T cells (*June et al., 2014*; *Gross and Eshhar, 2016*) and has been proven remarkably successful in treating liquid tumors.

However, the harsh microenvironment characterizing solid tumors manifests a major limitation to T-cell-based treatments. Hypoxic and acidic conditions at tumor sites, along with immunosuppressive capacity of tumor cells limit T cell infiltration, survival, and cytotoxicity (*Joyce and Fearon, 2015*; *Hanley and Thomas, 2020*; *Lim et al., 2020*). In contrast to T cells, myeloid cells migrate efficiently into the tumor mass and are well adapted to survive and function under this harsh environment (*Bruni et al., 2020*; *Greten and Grivennikov, 2019*). However, while myeloid cells are potentially able to produce cytotoxic compounds, they acquire an immunosuppressive phenotype once in the vicinity of the tumor cells and promote tumor growth (*Grivennikov et al., 2010*; *Mantovani and Sica, 2010*). Hence, equipping myeloid cells with means to release cytotoxic compounds following recognition of tumor antigens may provide a new therapeutic strategy. While recognition of tumors can be facilitated through scFv, which signaling chains should be fused are not clear. Early attempts to generate CAR myeloid cells have fused to CD3-$\zeta$ chain, which may function in these cells as Fc$\gamma$ chain (*Klichinsky et al., 2020*; *Sloas et al., 2021*). Yet, what signaling cascades should be induced in myeloid cells to elicit their cytotoxicity has remained unclear.

Here, we found that activation of the IgM receptor signaling in myeloid cells induces massive killing of tumor cells. However, we found that in contrast to lymphoid cells, which can ectopically express scFv, myeloid cells identify it as a misfolded protein and rapidly degrade it. Instead, we modified the high-affinity Fc$\gamma$RI, which normally binds IgG, to transmit IgM receptor-driven signals, while recognition of tumor antigens is mediated by IgG-tumor-binding antibodies. Indeed, incubation of mRNA-engineered myeloid cells with tumor cells resulted in massive tumor cell killing only in the presence of IgG-tumor-binding antibodies. Overall, this work suggests a novel mean to endow myeloid cells with antigen-specific killing abilities and harness their capacity to migrate and survive in the harsh conditions at the tumor microenvironment.

## Results

### IgM-induced signaling elicits cytotoxic response in macrophages and can be integrated to a CAR design

We have previously demonstrated that MHC-matched allogeneic tumors, which spontaneously regressed, are coated with IgG and IgM antibodies soon after tumor initiation and are found in proximity to tumor-infiltrating myeloid cells (*Carmi et al., 2015*). While we characterized the therapeutic role of tumor-binding IgG and their interactions with dendritic cells (DC; *Carmi et al., 2016*), the role of IgM in facilitating DC-mediated immunity has remained unclear. To address that, we initially compared their capacity to induce tumor immunity in a prophylactic tumor immunization assay. To this end, we incubated monocyte-derived dendritic cells (MoDC) with immune complexes (IC) composed of B16F10 tumor cells coated with allogenic IgG or IgM and injected them subcutaneously (s.c) to syngeneic mice (illustrated in *Figure 1A*). After two rounds of immunization, five days apart, mice were challenged with B16F10 cells and tumor size was monitored over time. We found that incubation of MoDC with IgG-IC, but not with IgM-IC, induced T cell immunity and prevented tumor growth

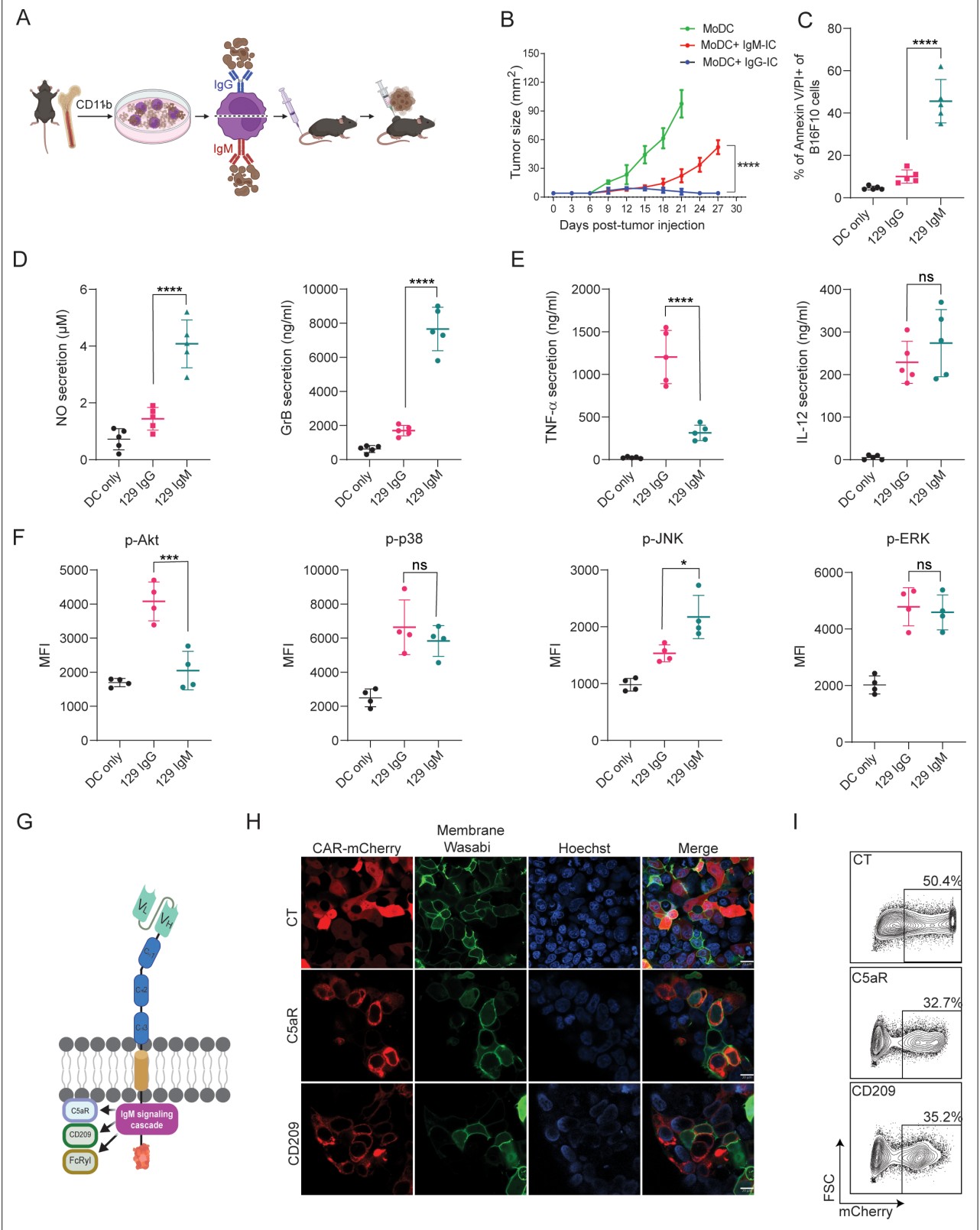

**Figure 1.** IgM-induced signaling elicits cytotoxic response in macrophages and can be integrated to a CAR design. (**A**) Illustration of experimental setting. (**B**) B16F10 tumor size (mm²) in mice following prophylactic immunization with MoDC pulsed with tumor cells coated with allogeneic IgG or IgM (n=4). (**C**) Mean percentages of B16F10 melanoma cells stained for Annexin V/PI incubated with allogenic IgG and IgM following incubation with MoDC (n=5). (**D–E**) Mean levels of Granzyme B and NO (**D**) and proinflammatory cytokines (**E**) in the supernatants of MoDC following overnight activation with

*Figure 1 continued on next page*

Figure 1 continued

IgG and IgM immune complexes (n=5). (**F**) Mean fluorescent intensity (MFI) of MAPK enzymes in MoDC following activation for 20 min with IgG and IgM tumor immune complexes (n=5). (**G**) Illustration representing CAR-macrophage design. (**H**) Confocal microscopy images of HEK293FT cells 24 hr post-transfection with CAR plasmids and membranous wasabi. (**I**) Representative FACS analysis of HEK239FT cells 24 hr post-transfection with CAR plasmids. Results are from one representative experiment out of at least three performed. Statistical significance was calculated using non-parametric t-test (* denote p<0.05, *** denote p<0.001, **** denote p<0.0001).

The online version of this article includes the following source data for figure 1:

**Source data 1.** IgM-induced signaling elicits cytotoxic response in macrophages and can be integrated to a CAR design.

---

(*Figure 1B*). We noticed however, that incubation in vitro of MoDC with tumor cells coated with allogenic IgM but not with IgG, resulted in a massive killing of the tumor cells (*Figure 1C*). To assess the underlying killing mechanisms, we measured the levels of nitric oxide (NO) and granzyme B (GrB) in the supernatants of overnight cultures. Consistent with their killing rates, significantly higher levels of both GrB and NO were detected upon incubation with allogenic IgM-IC (*Figure 1D*), indicating that both secretion of reactive oxygen species and lysosome deposition are involved in the killing. To further assess what signaling cascade is induced by IgM-IC, we tested the levels of major inflammatory cytokines and phosphorylated enzymes. Activation with IgG-IC induced classical singling through Fcγ receptors, characterized by high levels of TNFα and IL12, and activation of enzymes in the MAP kinase pathway (*Figure 1E, F*). The phenotype of IgM-IC activated MoDC was more complicated and characterized by low levels of TNFα and high IL-12, suggesting that Akt was only partially phosphorylated. Flow cytometric analysis indicated only neglectable levels of Akt phosphorylation, while ERK and p38 phosphorylation was comparable to that induced by IgG-IC along with higher levels of phospho-JNK (*Figure 1E, F*). This somewhat unique signaling cascade most likely indicates a combination of several signaling chains recruited by IgM, including Fcγ and C1q receptors.

To harness this signaling cascades to induce tumor cytotoxicity, we designed Chimeric Antigen Receptors (CAR) in which the antigen recognition region, a single chain fragment variable (scFv) derived from TA99 antibody, targets the melanoma antigen gp75, as previously reported (*Ma et al., 2019*). The scFv, consisting of Variable light (V$_L$) and variable heavy (V$_H$) chains was attached to a spacer region of Constant Heavy subunits 1–3 (CH$_{1-3}$). These subunits were fused to several potential IgM signaling chains in the transmembrane and intracellular regions and fused to a fluorescence tag to aid detection of expression (*Figure 1G*). Next, we transfected HEK 293 FT cells with the CAR constructs to test their expressions and cellular localization using confocal microscopy and flow cytometry. Indeed, all constructs were expressed on cell membrane within 24 hr (*Figure 1H, I*).

## scFv is not expressed by myeloid cells

Next, we sought to test the construct's expressions and functionality in myeloid cells. To this end, we transfected two established murine cell lines of dendritic cells (DC2.4) and macrophages (RAW 264.7) with our three constructs and with mCherry as a control plasmid. In both cell lines, about 30% of the cells expressed mCherry. In sharp contrast, however, we were unable to detect any construct expression in myeloid cells by either confocal microscopy or flow cytometric analysis (*Figure 2A–C*, *Figure 2—figure supplement 1A*). To test if this phenomenon would also occur in human cells, we transfected THP-1 monocyte cell line in comparison to Jurkat T cell line. Consistent with our results using mouse cells, we could not detect any CAR expression in THP-1 cells, while in Jurakt cells their expression was comparable to mCherry transfection rates (*Figure 2D, E*).

Since we could not detect expression of our constructs in myeloid cells, we sought to characterize the subunit leading to inhibition of expression by generating various constructs, in which different segments of CAR are omitted. Since myeloid cells were reported to express TRM21 E3 ligases that bind C$_H$1 and C$_H$2 fragments of internalized IgG (*Mallery et al., 2010*), we next generated a CAR construct with a minimal spacer region by elimination of Constant Heavy domain 1 and 2 (C$_H$1, C$_H$2) and tested the expression of these constructs in DC2.4 cell line. However, consistent with our previous results, we were unable to detect any protein expression (*Figure 2—figure supplement 1B–1C*). We then removed the rest of the spacers, generating a construct consisting only of the scFv region and an intracellular signaling domain. As with our previous finding, no expression in myeloid cell lines was

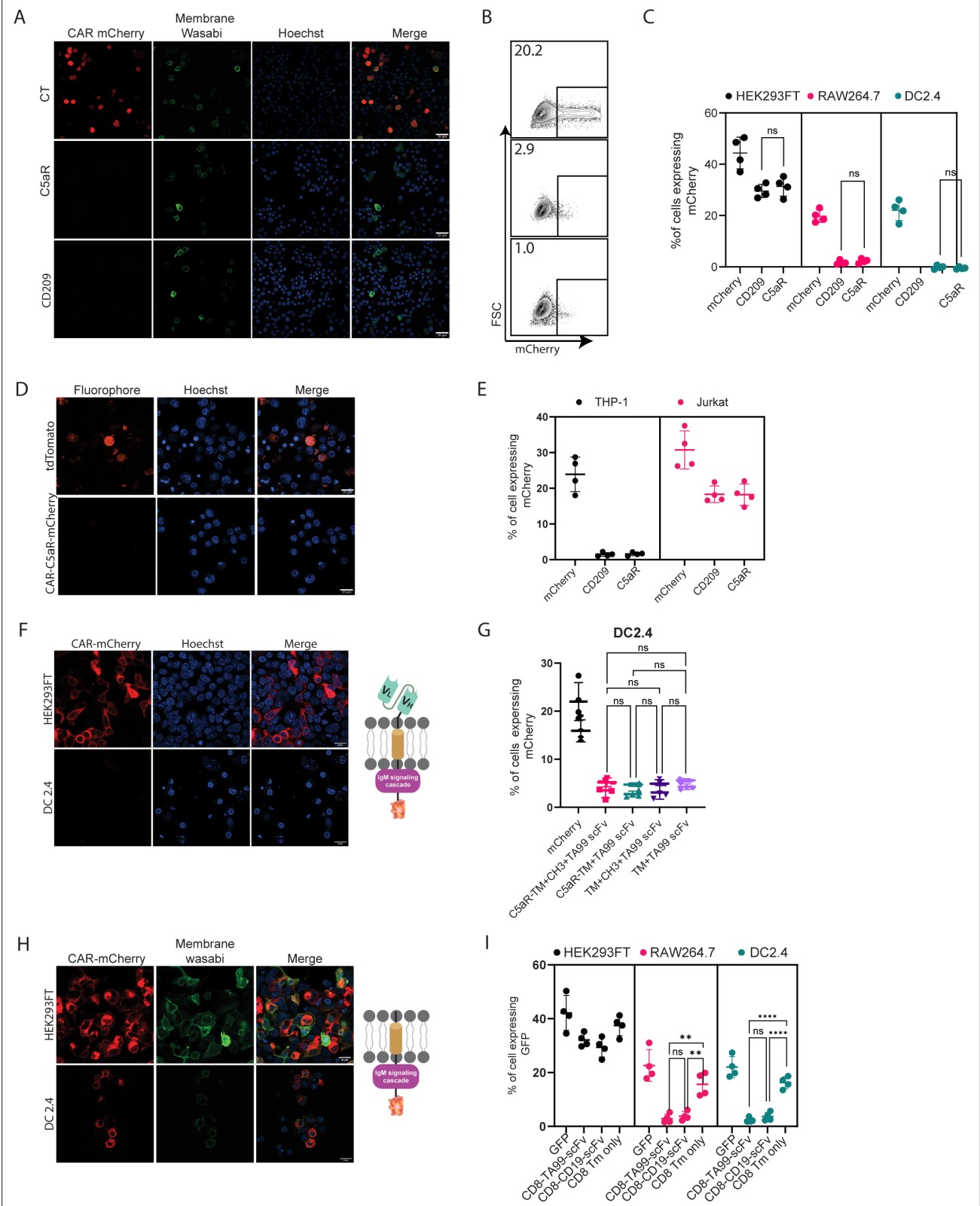

**Figure 2.** scFv is not expressed by myeloid cells. (**A**) Confocal microscopy images of DC 2.4 cells 24 hr post-transfection with CAR plasmids and membranous wasabi. (**B**) Representative FACS analysis of DC 2.4 cells 24 hr post-transfection with CAR plasmids. (**C**) Percentages of transfected cells 24 hr post-transfection (n=4). (**D**) Confocal microscopy images of THP-1 cells 72 hr post lentiviral infection with CAR-C5aR-mCherry and tdTomato plasmids. (**E**) Percentages of transfected human cell lines 72 hr following transduction (n=4). (**F–G**) Representative confocal microscopy (**F**) and mean

*Figure 2 continued on next page*

*Figure 2 continued*

percentages (**G**) of cells expressing chimeric molecules 24 hr after transfection. (**H–I**) Representative confocal microscopy (**H**) and mean percentages of cells (**I**) expressing chimeric molecules 24 hr following transfection (n=4). Results are from one representative experiment out of at least three performed. Statistical significance was calculated using non-parametric t-test (** denote p<0.01, **** denote p<0.0001).

The online version of this article includes the following figure supplement(s) for figure 2:

**Figure supplement 1.** scFv is not expressed by myeloid cells.

detected (*Figure 2F, G*). Similarly, removal of the intracellular portion did not alter the expression patterns of TA99-derived scFv, suggesting that the scFv domains may be the element that prevents its expression (*Figure 2G, H*). To verify that these expression patterns do not stem from the use of mCherry, we have also fused these constructs to GFP. Nonetheless, in both cases, the presence of scFv prevented protein expression in macrophages and DC (*Figure 2—figure supplement 1C*).

To test if this phenomenon is restricted to TA99 CAR or reflects a broader phenomenon, we sought to compare TA99 CAR to a well-established αCD19-derived scFv. Towards this end, we generated a plasmid that contains the CD8 signal peptide followed by αCD19-derived scFv and fused to CD8 spacer and transmembrane portion, and an identical one that lacks the scFv part (*Figure 2H*). Interestingly, while we detected low levels of αCD19-derived scFv, they were localized in intracellular compartments and not on the cell membrane (*Figure 2—figure supplement 1D*). Removal of scFv domain enabled expression of the inserted protein in the Golgi and cell membrane of both macrophages and DC cells (*Figure 2H, I*, *Figure 2—figure supplement 1E*).

## Both VH and VL domains prevent the expression of scFv in myeloid cells

We next aimed to test which portion of scFv prevents its membranal expression. Initially, we designed two plasmids in which all other signaling subunits were removed, leaving only the TA99- or αCD19-derived scFv fragment. As with our previous findings, here too we were not able to detect intracellular TA99- nor αCD19-derived scFv expression (*Figure 3A, B*). To test the possibility that the location of the fluorescent tag inhibits scFv expression, we transfected DC2.4 with a GFP fused to the C-terminus of αCD19-derived scFv. Similarly, this protein did not lead to scFv expression (*Figure 3—figure supplement 1A*). Next, we generated plasmids consisting of minimal subunits, consisting of only the variable light or variable heavy domains of TA99 and αCD19. While expression levels were somewhat higher in these constructs, compared to the full length of scFv, they were still benign and significantly lower compared to GFP only (*Figure 3C–E*).

As a result, we sought to further investigate whether the primary protein structure of amino acid sequence or the tertiary immunoglobulin structure of the scFv subunits prevent the expression. To this end, we inserted point mutations that switched cysteine amino acids to glycine, creating a linear protein by preventing disulfide bonds. Interestingly, linearization of αCD19 variable light chain (or heavy chain) had only a marginal effect on its expression patterns and resulted in reduced and compartmentalized expressions (*Figure 3F–H*). In addition, we generated a plasmid that consisted of the first and last 15 amino acids of variable light chain which includes the cysteine amino acids for disulfide bond formation. This fragment was almost inert, and its expression patterns were comparable to that of GFP only (*Figure 3—figure supplement 1B*). We also tested whether specific fragments within the variable light chain sequence prevented its expression. Dividing the light chains of αCD19 scFv into three sequences resulted in a high expression in myeloid cells, albeit tending to induce different expression patterns of cellular compartmentalization (*Figure 3G–H*). Taken jointly, our results suggest that the linear form of either the variable light or the heavy chain of antibodies results in a poor and compartmentalized protein expression compared to the full length of ScFv to a lesser extent.

## scFv induce ER stress in myeloid cells

To elucidate the level at which scFv constructs are silenced, we sought to transfect cells of myeloid lineage with mRNA composed of αCD19 scFv fused to a green fluorescent protein tag. Nonetheless, this form of transfection did not lead to expression of scFv in myeloid cells (*Figure 4A*). Additionally,

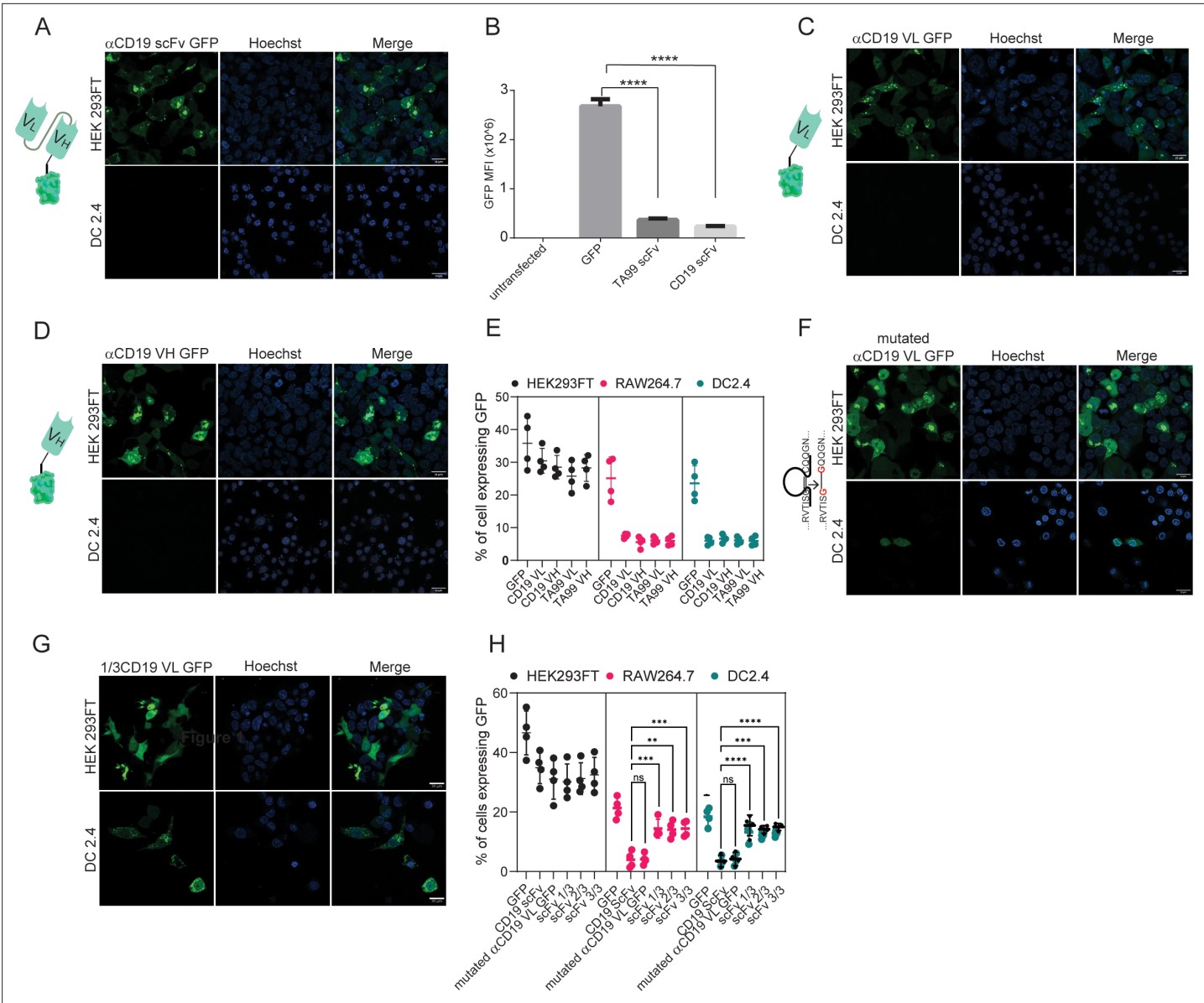

**Figure 3.** Both VH and VL domains prevent expression of ScFv in myeloid cells. (**A**) Confocal microscopy images of DC 2.4 cells 24 hr post-transfection with αCD19-scFv GFP plasmid. (**B**) Geometric mean of GFP-positive cells 24 hr post-transfection with different αCD19- and TA99- ScFv GFP constructs in DC2.4 (n=3). (**C**) Confocal microscopy images of HEK 293 FT and DC 2.4 cells 24 hr post-transfection with αCD19-variable light chain GFP plasmid. (**D**) Confocal microscopy images of HEK 293 FT and DC 2.4 cells 24 hr post-transfection with αCD19-variable heavy chain GFP plasmid. (**E**) Mean percentages of cells expressing scFv fragments 24 hr post-transfection (n=4). (**F**) Left: Illustration of mutated variable light chain. Right: Confocal microscopy images of HEK 293 FT and DC 2.4 cells 24 hr post-transfection with αCD19- mutated (linear) variable light chain. (**G**) Confocal microscopy images of HEK293FT and DC2.4 cells 24 hr post-transfection with 1/3 fragments of αCD19-variable light chain GFP plasmid. (**H**) Mean percentages of GFP-positive cells 24 hr post-transfection with different fragments of scFv-GFP in DC2.4 and RAW264.7 (n=4). Results are from one representative experiment out of at least three performed. Statistical significance was calculated using non-parametric t test (** denote $p<0.01$, *** denote $p<0.001$, **** denote $p<0.0001$).

The online version of this article includes the following figure supplement(s) for figure 3:

**Figure supplement 1.** Immunoglobulin structure of scFv does not prevent degradation by myeloid cells.

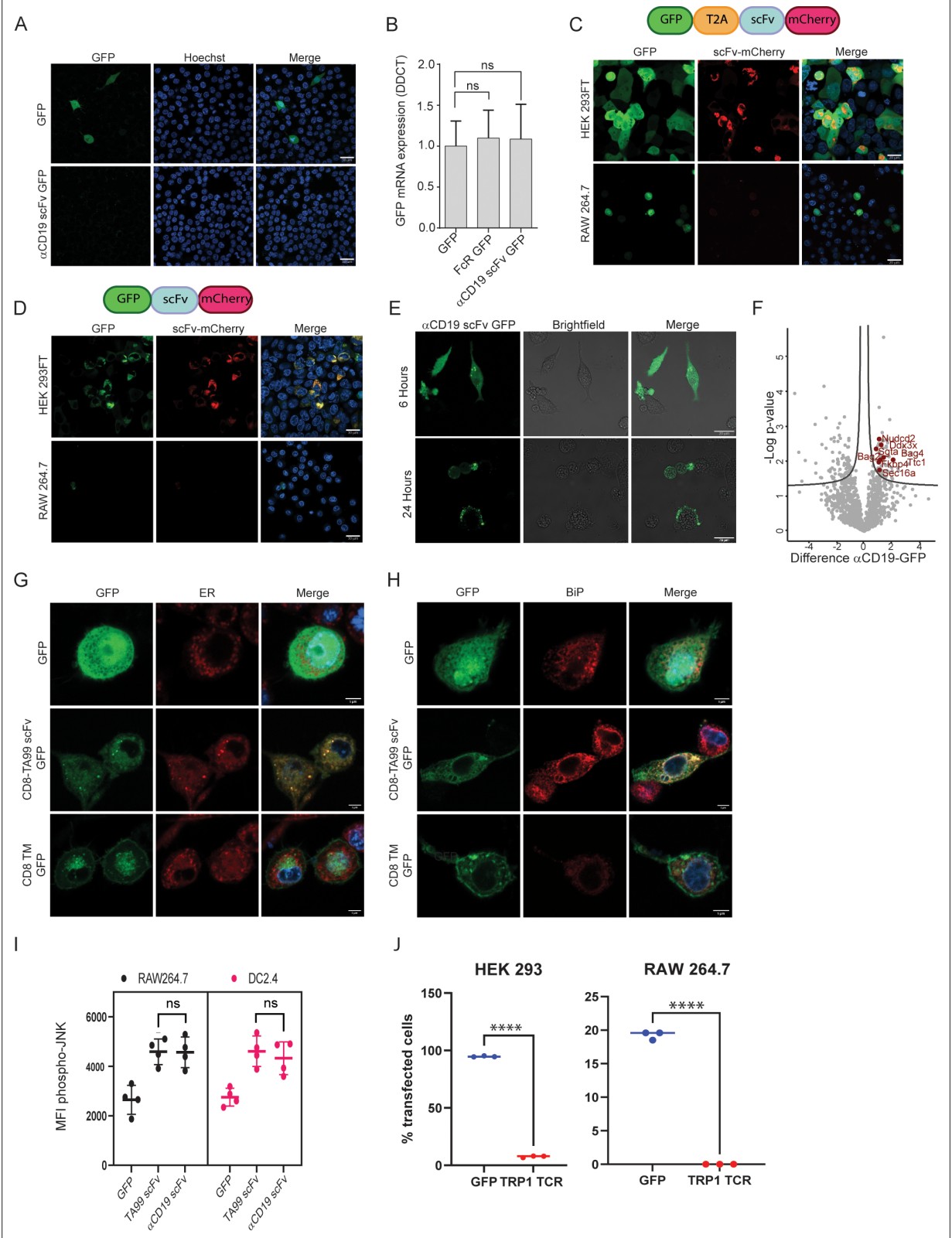

**Figure 4.** scFv fragments induce ER stress in myeloid cells. (**A**) Confocal microscopy imaging of RAW 264.7 24 hr post-transfection with linear mRNA vectors translating to GFP and αCD19-scFv GFP. (**B**) qPCR data showing relative mRNA levels in RAW 264.7 transfected with GFP, Fc receptor-GFP, and αCD19-scFv GFP (n=4). (**C**) Upper: Illustration of plasmid subunits. Lower: Confocal microscopy imaging of HEK293FT and RAW 264.7 24 hr post-transfection with T2A ribosomal skipping plasmid including ScFv. (**D**) Upper: Illustration of plasmid subunits. Lower: Confocal microscopy imaging of

*Figure 4 continued on next page*

*Figure 4 continued*

HEK293FT and RAW 264.7 24 hr post-transfection with plasmid not containing T2A. (**E**) Confocal microscopy images of RAW264.7 cells at 6 hr and 24 hr post-transfection with αCD19-ScFv GFP plasmid. (**F**) Volcano plot showing differentially expressed proteins αCD19 ScFv GFP and GFP in DC 2.4 cells. (**G**) Confocal microscopy images of DC 2.4 stained with an ER stain, 24 hr post-transfection with GFP, membranous TA99-ScFv GFP. (**H**) Confocal microscopy images of DC 2.4 stained for BiP 24 hr post-transfection. (**I**) Mean levels of phospho-JNK 6 hours following transfection (n=4). (**J**) Percentage of cells expressing GFP or TRP-TCR1 24 hr following transfection (n=3). Results are from one representative experiment out of at least three performed. Statistical significance was calculated using non-parametric t-test (**** denote p<0.0001).

The online version of this article includes the following source data and figure supplement(s) for figure 4:

**Source data 1.** scFv fragments induce ER stress in myeloid cells.

**Figure supplement 1.** scFv induces ER stress response in myeloid cells.

we measured the mRNA levels of GFP mRNA in mouse myeloid cells following transfection with αCD19 scFv-GFP vectors, in comparison to GFP only or to Fc receptor fused to GFP (which are robustly expressed in these cells). Similar levels of GFP mRNA were detected in all transfected cells (*Figure 4B*). Based on these findings, we speculate that the negative regulation on scFv expression is not evident at the mRNA level.

To further assess this possibility, we utilized 2 A ribosomal skipping peptide in order to test if scFv is regulated at the post-translational level. Hence, we generated plasmids containing GFP fluorescence tag followed by T2A skipping peptide, succeeded with TA99 scFv fused to mCherry, all in one reading frame. This design allowed us to test if the GFP and scFv expression are coupled, thus indicating that the regulation is made on the mRNA transcript and prior to protein translation. Alternatively, expression of one fluorescent tag would indicate that the regulation is made at the protein level following translation. While we detected both tags (GFP, mCherry) in HEK293 cells, only GFP was expressed in myeloid cells, but not the mCherry which was fused to scFv (*Figure 4C*). Eliminating the T2A skipping peptide from this construct, generating a plasmid containing GFP tag fused to TA99 scFv succeeded by mCherry, resulted in abrogated and reduced expression levels of both fluorescent tags in myeloid cells (*Figure 4D*). These results therefore suggest that scFv expression is regulated on the protein level.

Consistent with this notion, we found that scFv is expressed in myeloid cell at earlier time points, starting at four hours post-transfection and peaking at about 6 hr (*Figure 4E*). In order to determine the mechanism through which scFv is degraded, we next performed proteomics of immunoprecipitated scFv. Hence, macrophages were transfected with GFP only, or GFP fused to αCD19 scFv. After 6 hr, GFP-expressing cells were sorted and lysed, and associated proteins were pulled down using anti-GFP conjugated beads. Eluted proteins were then sent to analyses by mass spectrometry (*Figure 4—figure supplement 1A*). Analysis of protein interactors with scFv in myeloid cells indicated that scFv fraction was statistically enriched with proteins associated with ER stress that could promote scFv degradation (*Figure 4F*). To further corroborate that possibility, we assessed the co-localization of scFv with key proteins in this pathway. Confocal microscopy corroborated that the majority of αCD19 scFv were localized at the endoplasmic reticulum (*Figure 4G*) and co-localized with BIP (*Figure 4H*) and G3BP1 (*Figure 4—figure supplement 1B*), which are master regulators of ER stress. Lastly, flow cytometric analysis indicated higher levels of phospho-JNK (*Figure 4I*). Next, we tested if TCR fragments, which have been shown to induce ER stress in cells lacking the corresponding chaperons for their folding, will be expressed in myeloid cells. Consistent with this notion, we could not detect expression of alpha chain from TRP1-reactive TCR *Dougan et al., 2013* following transfection of RAW264.7 cells (*Figure 4J* and *Figure 4—figure supplement 1C*). Lastly, we assessed whether scFv fragments are presented on MHC-I and MHC-II. To this end, we transfected Raw 265.7 cells with MHC-I peptide OVA$_{257-264}$ followed by MHC-II OVA$_{323-339}$ peptide alone or fused to the C-terminal of αCD19 scFv. After 16 hr, we incubated T cells from OT-I and OT-II mice and tested their rates of proliferation (*Figure 4—figure supplement 1D*). Yet, only negligible differences were observed in T cells incubated with RAW264 cells expressing the different constructs (*Figure 4—figure supplement 1E*).

# FcγRI provides a scaffold for incorporating IgM-induced signaling in myeloid cells and endows them with tumor-cell-specific killing ability

Taking our findings into consideration, we sought to adopt an alternative approach; instead of equipping monocytes with scFv segment, we fused the α chain of the high-affinity Fc IgG receptor (FcγRI) as extracellular region fused to signaling chain (*Figure 5A*). These construct's recognition of the target cells was designed to be mediated by tumor-binding antibodies. First, we tested the expression levels of the naive and the chimeric FcγRI constructs in RAW264.7 cells. Flow cytometric analysis 24 hr post-transfection indicated that both constructs were successfully expressed at comparable levels (*Figure 5B, C*). Confocal microscopy further indicated that both constructs expressed on the cell membrane and Golgi. (*Figure 5D*). Similar expression patterns were also observed in infected murine bone marrow cells (*Figure 5E*).

Consequently, we sought to characterize the effect of tumor-binding antibodies on macrophage cytotoxic activity. To this end, we co-incubated tdTomato-labeled 4T1 tumor cell lines expressing human HER2$^+$ with RAW264.7 cells that constitutively express the chimeric receptors with and without the anti-HER2$^+$ antibody *trastuzumab*. Confocal microscopy indicated that RAW264.7 alone had no cytotoxic activity following a 24 hr incubation. In contrast, the addition of *trastuzumab* induced a polarized accumulation of GrB in the synapse between macrophages and tumor cells. Significantly higher levels of GrB in the immune synapse were counted in cells expressing C5aR chimeric receptor (*Figure 5F, G* and *Figure 5—figure supplement 1A*). To quantify the killing capacity of this construct, we next transfected RAW264.7 macrophages with mRNA which encode these modified receptors, and incubated them with fluorescently labeled HER2$^+$ expressing 4T1 tumor cells under IncuCyte. Consistent with GrB polarization, transfected macrophages alone were almost completely inert whereas the addition of *trastuzumab* promoted tumor cell killing (*Figure 5H*). Importantly, while killing rates of macrophages expressing C5aR chimeric receptor were higher, they were also induced by free and irrelevant antibody (e.g. rituximab; *Figure 5H*).

To assess the underlying mechanism, we analyzed the transfected RAW264.7 cells under high-resolution microscopy. In steady state, cells transfected with the native FcγRI had a clear separation between the alpha chain (antibody binding chain) and their signaling gamma chain. The addition of free *trastuzumab* did not alter their membranal architecture. Incubation with plastic-immobilized *trastuzumab* resulted in polarization of the alpha chain toward the plastic and co-localization with the gamma chain. The architecture of C5aR chimeric receptor completely abrogated the membrane organization of the signaling chains, as the two co-localized even under steady state (*Figure 5I*, *Figure 5—figure supplement 1B*). We then speculated that the fusion of such signaling chain to the alpha chain could mitigate the effects of tonic signaling by attaching IgM receptor signaling chain, instead of the gamma chain, to the transmembrane domain (illustrated in *Figure 5—figure supplement 1C*). Indeed, this architecture can be highly expressed in macrophage membrane and did not recruit or abrogate the gamma chain upon addition of antibodies, while maintaining alpha chain clustering to immobilized *trastuzumab* (*Figure 5—figure supplement 1D–1F*). This construct promotes polarized expression of GrB at the immunological synapse and induces an antibody-specific tumor cell lysis (*Figure 5J*, *Figure 5—figure supplement 1G*). Lastly, we tested whether myeloid cells expressing modified FcγRI can affect tumor growth in vivo. To this end, mice were challenged with B16F10 tumor cells and to grow for 8 days until they reached an average size of 20 mm$^2$. Next, mRNA transfect BMDC were injected s.c. once a week alone or in combination with i.p. injections of TA99 antibody against the melanoma antigen TRP1. Indeed, significant inhibition of tumor growth was observed in mice treated with modified FcγRI and TA99, but not with sham transfected BMDC or without TA99 antibodies (*Figure 5K* and *Figure 5—figure supplement 1H–1I*). Analysis of immune cell infiltration indicated higher rates of T cells in tumors treated with modified DC and TA99. (*Figure 5L*) suggesting this treatment also potentiate the hosts immune response.

## Discussion

The microenvironment of most solid tumors is often enriched with myeloid cells at different stages of maturity and polarization and the general notion suggests that they often adopt a suppressive phenotype that supports tumor progression and escape (*Bruni et al., 2020*; *Mantovani and Sica, 2010*; *Gajewski et al., 2013*). The inherent plasticity of these cells, however, enables them to change their

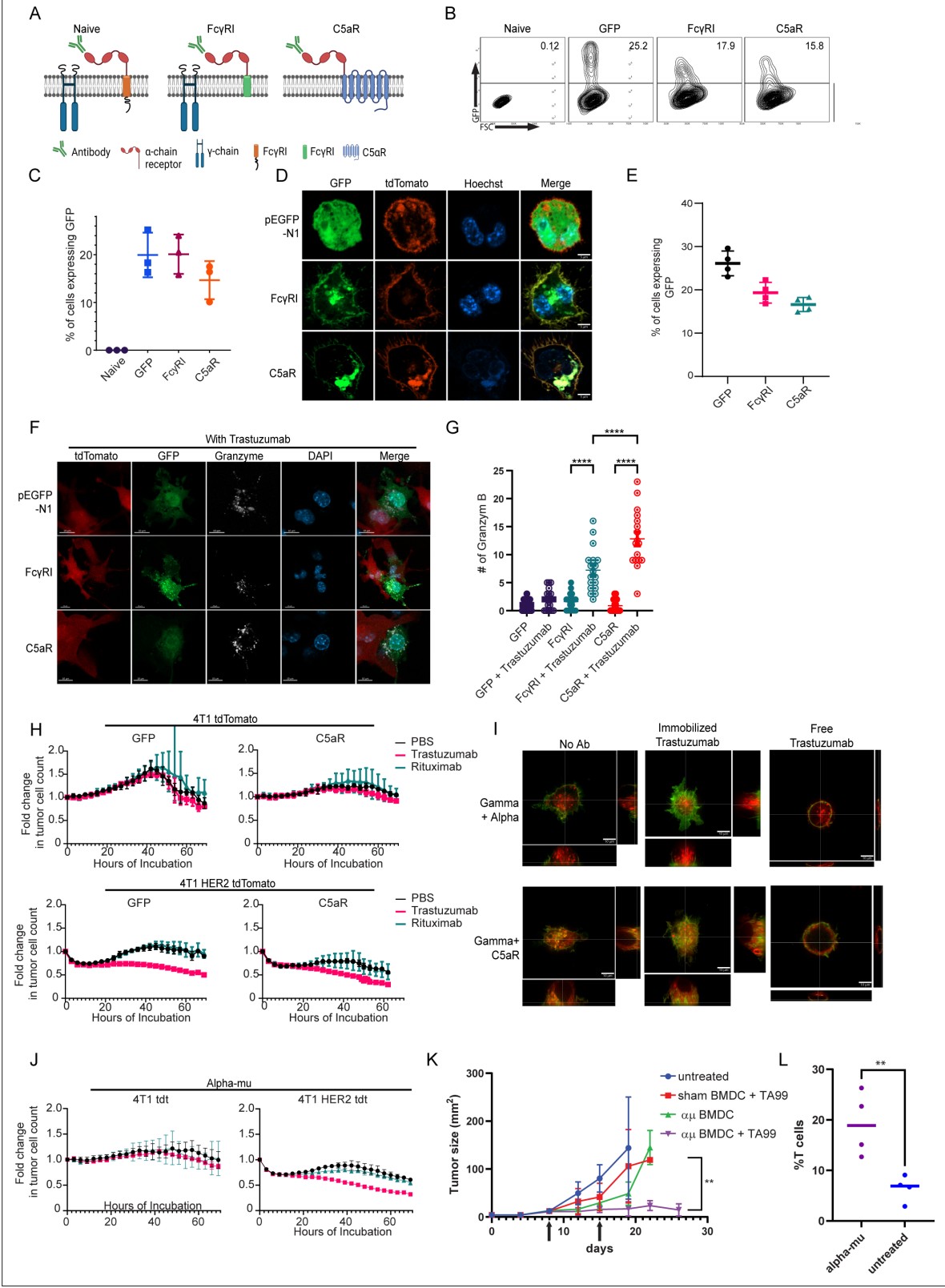

**Figure 5.** FcγRI can provide a scaffold for incorporating IgM-induced signaling in myeloid cells and endows them with tumor cell-specific killing ability. (**A**) Illustration of chimeric Fcγreceptor design. (**B–C**) Representative FACS plots (**B**) and mean percentages (**C**) of RAW 264.7 cells expressing chimeric Fcγ receptors 24 hr after transfection (n=3). (**D**) Confocal microscopy images of RAW 264.7 cells 24 hr post-transfection with Fcγ receptors tagged with GFP and membrane-tagged tdTomato. (**E**) Mean percentages of BMDC 72 hr post lentivirus transduction with Fcγ receptors (n=4). (**F**) Confocal

*Figure 5 continued on next page*

*Figure 5 continued*

microscopy staining of GrB in RAW 264.7 cells co-cultured overnight with 4T1 cells expressing human HER2. (**G**) Mean counts of GrB in the synapse between transduced RAW 264.7 cells and the tumor cells (n=18). (**H**) IncuCyte analysis of human HER2$^+$ 4T1 cells growth following incubation with transduced RAW 264.7 cells (n=6). (**I**) Super-resolution microscopy of GFP-tagged chimeric FcγR and mCherry-tagged gamma chain. (**J**) IncuCyte analysis of human HER2$^+$ 4T1 cells growth following incubation with transduced RAW 264.7 cells (n=6). (**K**) Tumor size measurements (mm$^2$) in mice treated with mRNA transfected BMDC with and without antibody. Arrows point to subcutaneous injection of treated BMDC (n=5). (**L**) Mean percentages of CD3$^+$ out of CD45$^+$ cells in B16F10 tumors from day 26 (n=4). Results are from one representative experiment out of at least three performed. Statistical significance was calculated using non-parametric t test (** denote $p < 0.01$, **** denote $p < 0.0001$).

The online version of this article includes the following source data and figure supplement(s) for figure 5:

**Source data 1.** FcγRI can provide a scaffold for incorporating IgM-induced signaling in myeloid cells and endows them with tumor cell-specific killing ability.

**Figure supplement 1.** FcγRI can be used as a scaffold to transmit IgM-induced signaling.

phenotype upon injection of adjuvant or immune-stimulatory agents (*Quatromoni and Eruslanov, 2012*; *Ricketts et al., 2021*; *Bercovici et al., 2019*). Countless experimental and clinical data have been tested for their capacity to convert tumor-infiltrating myeloid cells phenotype to an anti-tumor, including TLR agonists (*Cheng et al., 2020*; *Dudek et al., 2007*; *Miura et al., 2016*; *Rodell et al., 2018*), immune complexes (*Carmi et al., 2015*; *Gao et al., 2018*; *Ackerman et al., 2021*), immune checkpoints (*Arlauckas et al., 2017*; *Gordon et al., 2017*) and co-stimulatory antibodies (*Beatty et al., 2011*). While they showed improved T cell infiltration and overall lower tumor burden, they did not meet the expectations. It is noteworthy that most of these attempts are based on the notion that CD8$^+$ T cells are the main effector arm against cancer and that the purpose of myeloid cell activation is to support T cell-mediated killing (*Ricketts et al., 2021*; *Bercovici et al., 2019*). Nonetheless, myeloid cells are highly equipped with cell-killing mechanisms, including antibody-medicated cellular toxicity (ADCC) and secretion of reactive oxygen species (*Mantovani et al., 2017*). Here, we demonstrated that activating the IgM-receptor signaling in myeloid cells using tumor-binding IgG induces oxidative burst, Granzyme release, and massive tumor cell lysis.

While genetic engineering of immune cells to treat cancer has proven to be successful in lymphoid cells (*Gross and Eshhar, 2016*; *Kalos et al., 2011*), cells of the myeloid lineage resist gene modification. Noteworthily, these cells are traditionally thought to be programmed to respond to foreign genetic materials. Early attempts to overcome this inherent barrier were achieved by Biglari et al. using adenoviral vectors. They demonstrated that the fusion of scFv against CEA to FcγR can be expressed in monocytes (*Biglari et al., 2006*). Although this construct leads to granzyme release in T cells, its function in monocytes remains obscure. Further attempts to equip macrophages with specific tumor antigen recognition were done by Morrissey et al. They fused scFv derived from αCD19 antibody to different signaling chains that facilitate phagocytosis (*Morrissey et al., 2018*). Interestingly, most of these receptors do not appear to be expressed on the cell membrane. Moreover, CAR-merTK, which appears on the cell membrane, was functionally inert and did not induce phagocytosis.

Recently, Klichinsky et al. have attempted to genetically modify macrophages using an adenoviral vector Ad5f35 expressing αCD19 and αHER2 Chimeric Antigen Receptor fused to CD3 ζ signaling domain. The authors suggested that CAR expression and activation in human macrophages could be employed as a strategy to kill tumor cells expressing the corresponding antigens through phagocytosis (*Klichinsky et al., 2020*). It may be that the use of Adenovirus as transduction vectors, or human macrophages cultured from blood monocytes enable overcoming the limitations described in the present manuscript.

Why scFv induces ER stress specifically in myeloid cells remains unclear. Activation of ER stress by misfolded proteins in macrophages plays a dominant role in many human pathologies, yet it also occurs in other cell types. One possible explanation is that these cells lack the machinery needed for successful folding of antibody-like structures. Along these lines, many non-immune cells such as HeLa or HEK293 cells initiate ER-stress response upon transfection of fragments derived from immune-associated receptors such as the alpha chain of the T cell receptor (*Oslowski and Urano, 2011*).

Overall, while we identify FcγRI as a possible construct that could endow myeloid cells with tumor-specific recognition abilities, the main goal of the present work is to highlight the unique challenges

and considerations that are involved in genetically reprogramming the signaling in myeloid cells. We demonstrate that myeloid cells cannot express scFv, most likely since they lack the corresponding chaperons required for its folding. Yet, more work is needed before this could be promoted as an efficient therapy for solid tumors.

## Materials and methods

### Mice

Wild-type (WT) C57BL/6 mice were purchased from Envigo (Jerusalem, Israel). 129S1 mice were purchased from Jackson laboratories (Bar Harbor, Maine, USA) C57BL/6-Tg(TcraTcrb)1100Mjb/J OT-I mice were kindly gifted from Prof. Carmit Levy, Tel Aviv University, CD45.1 congenic mice were kindly gifted from Prof. Adi Barezel, Tel Aviv University, and B6.Cg-Tg(TcraTcrb)425Cbn/J OT-II mice were generously gifted from Prof. Jakub Abramson, Weizmann Institute, and were housed and maintained in a specific pathogen-free (SPF) conditions animal facility in Tel-Aviv University according to the American Association for the Accreditation of Laboratory Animal. Male and female 8–12 weeks-old mice were used in all experiments. Animal experiments were approved by the Tel-Aviv University ethics committee.

### Cell lines

Human Embryonic Kidney (HEK)–293 FT cells were purchased from Thermo Fisher Scientific (Waltham, MA, USA). DC 2.4 were a kind gift from Dr. Kenneth Rock Gross from Umass cHAN Medical School. RAW 264.7 and Jurkat cell lines were purchased directly from ATCC. THP-1 were kind gifts from Prof. Mordechay (Motti) Gerlic from the Department of Clinical Microbiology and Immunology at Tel Aviv University. B16F10 (CRL-6475) cells and 4T1 (CRL-2539) cells were directly purchased from ATCC. Cells were cultured in DMEM supplemented with 10% heat-inactivated FBS, 2 mM L-glutamine, and 100 µg/mL penicillin/streptomycin (all from Biological Industries, Beit Haemek, Israel). All cells were routinely tested for mycoplasma (EZ-PCR Mycoplasma Test Kit, Biological Industries).

### Primary cells

For bone-marrow DC (BMDC), C57BL/6 mice were sacrificed using a $CO_2$ chamber. Femurs, hip bones, and tibias were obtained and kept in DMEM supplemented with 10% FBS. Bones were washed twice with Phosphate Buffered Saline (Gibco, NY, USA) and sterilized for 10 s in a 70% ethanol solution. The cleaned bones were crushed with a sterile mortar and pestle in full medium and the cell mixture was filtered through a 40 µM cell strainer. Cells were then centrifuged and resuspended in complete DMEM supplemented with 50 ng/mL GM-CSF and 10 ng/m IL-4 (PeproTech, Rehovot, Israel) and plated at a concentration of $1x10^6$/mL for 6–7 days.

### Transient transfection

#### Liposomes-based transfection

For transfecting HEK293 cells, jetOPTIMUS reagent (Polyplus Transfection, Strasbourg, France) was used according to manufacturer protocol. For DNA transfection of macrophages, Lipofectamine 2000 (Invitrogen, Waltham, MA, USA) and jetPEI-Macrophage (Polyplus Transfection) were used.

#### Electroporation

$3x10^6$ cells were suspended in 0.25 mL Opti-MEM medium, mixed with the relevant plasmids, and placed on ice for 20 min in 4 mm cuvettes. Cell electroporation was performed using Gene Pulser Xcell electroporation system (Bio-Rad, Hercules, CA, USA). electroporation protocol: voltage - 250 V, capacitance 900 µF, resistance $\infty$ Ω. Immediately following pulsation, cells were washed with pre-warmed DMEM media, centrifuged, and plated in culture plates.

### Production of lentiviral particles

A total of $8×10^6$ HEK-293FT cells were plated on a 10 cm plate pre-coated with 200 µg/mL poly-L-lysine and left to adhere overnight and reach a confluence of 80%. pLVX plasmids containing either receptor of interest tagged with a fluorescent protein or wasabi/ tdTomato control under an EF1 promoter were mixed with psPAX2 (a gift from Didier Trono, Ecole Polytechnique Fédérale de Lausanne, Lausanne,

Switzerland; Addgene plasmid 12260) and pCMV-VSV-G (a gift from Bob Weinberg, Massachusetts Institute of Technology, Cambridge, Massachusetts, USA; Addgene plasmid 8454) at a molar ratio of 3:2:1, and cells were transfected using jetOPTIMUS reagent (Polyplus Transfection). After 24 hr, the medium was replaced with complete DMEM supplemented with 0.075% sodium bicarbonate (Biological Industries). Virus-containing medium was harvested after 24 hr and 48 hr. Following collection, the supernatant was passed through a 0.45 µm membrane filter to rid it of cellular debris. In order to produce high viral titer stock, viral-containing media was concentrated via centrifugation in Amicon Ultra-15 centrifugal filter unit (100 KDa; Sigma-Aldrich, St. Louis, MO, USA).

## Tumor cell lines viral infection

For transduction of tumor cells, cells were incubated with viruses and 100 µg/mL polybrene (Sigma Aldrich) for 30 min followed by 30 min of centrifugation before medium was replaced. Following three days, cells that expressed HER2$^+$/tdTomato were sorted by FACSAriaII.

## Mouse IgG and IgM purification

Mouse antibodies were obtained from pooled 5 mL mouse blood obtained from Inferior Vena Cava. Whole blood was left on ice for 20 min to allow blood to clot. Blood was then centrifuged at 600 RCF for 10 min and serum was collected and re-centrifuged at 20,000 RCF for an additional 10 min. Serum was filtered through 0.1 µM and total the IgG and IgM were purified using protein-G and 2-mercaptopyridine columns, respectively (GE Healthcare, Chicago, IL, USA). The levels of purified IgG and IgM were measured with specific ELISA kits (Bethyl, Montgomery, TX, USA) according to manufacturer's instructions.

## Antibody-tumor lysate immune complexes (Ig -IC) and antibody-bound tumor cells

When obtained from surgical resections, tumor cells were initially isolated after enzymatic digestion and sorted as FSC$^{hi}$/CD45$^{neg}$ cells prior to their fixation and staining. For tumor-antibody complexes, tumor cells were fixed in 2% paraformaldehyde, washed extensively, and incubated with 2–5 µg allogeneic IgG per 1x10$^5$ tumor cells, and then washed to remove excess antibodies. DC activation with the above Ig-IC was repeated in at least five independent experiments in biological duplicates.

## In vivo tumor vaccination models

For tumor recurrence studies, 2x10$^5$ tumor cells were injected s.c. above the right flank, and the size of growing tumors was measured using calipers. When tumors reached 16–25 mm$^2$ for B16F10 tumors, mice were anesthetized, and visible macroscopic tumor was surgically removed. Resected tumors were enzymatically digested with 0.1 mg/mL of DNase I (Sigma Aldrich) and 5 mg/mL collagenase IV (Sigma Aldrich) in HBSS. Cells were then fixed in 2% paraformaldehyde for 10 min, washed extensively in PBS, and coated for 30 min with syngeneic or allogeneic antibodies. Antibody-coated tumor cells were then washed and injected into tumor-resected mice (2x10$^6$ per mouse) or were added to DC cultures. After overnight incubation, DC were washed, and 2.5x10$^6$ were injected s.c. into tumor-resected mice one day after the tumors were removed, adjacent to the site of tumor resection. For prophylactic vaccination assays, blood, and tumor DC were incubated overnight with B16F10-IC, DC were then washed and 6x10$^6$ cells per mouse were injected s.c. After 5 days 2.5x10$^4$ B16F10 cells were injected s.c. above the right flank and tumor growth was monitored. Experiments were repeated independently at least three times with four to five mice per group.

For treatment of established melanoma, C57Bl/6 female mice bearing approximately 20 mm$^2$ B16F10 tumors were injected s.c. with 2x10$^6$ BMDC transfected with modified FcγRI and with i.p. injection 250 µg per mouse of TA99 antibody (BioXcell, Lebanon, NH, USA). Mice were treated twice, one week apart and tumor size was measured using a caliper twice a week.

## Kinase phosphorylation measurements

For measurement of kinase phosphorylation by FACS, 1x10$^5$ mouse MoDC were generated from C57Bl/6 bone marrow. Thereafter, 1x10$^5$ B16F10 tumor- IC were added to DC cultures for 1 min and 15 min Fluorochrome-conjugated antibodies against phospho-p38 (Thr180/Tyr182), phospho-JNK (Ser63), phospho-ERK1/2 (p44) (pT202/pY204) and phospho-Akt (pY473) were purchased from Cell

Signaling Technologies (Danvers, MA, USA). DC protein phosphorylation experiments were repeated 2–4 times.

## NO and granzyme B measurements

BMDC were activated with IC composed of B16F10 cells coated with either IgG or IgM isolated from the 129S1 mice. GrB was measured using mouse granzyme B ELISA kit (Abcam, Cambridge, UK) and NO was measured using Griess reagent system (Promega, Madison, WI, USA) according to manufacturer's protocol.

## Mass spectrometry

### Sample preparation

DC 2.4 were transfected with either αCD19 scFv-GFP or GFP plasmids using Lipofectamine 2000 transfection reagent. Four to 6 hr post-transfection, when scFv expression was at its maximal level during the expression window (determined by prior calibration experiments), cells were harvested and sorted based on GFP expression using BD FACSAria III Cell Sorter. GFP-positive cells collected were resuspended in cell lysis buffer based on PBS + protease inhibitor cocktail (NEB#5871) and lysis was performed using cell sonication (Sonics Vibra Cell VCX 130 Digital Ultrasonic Processor). Sonication protocol: time: 100 s, pulse on: 10 s, pulse off: 10 s, amplitude 25%.

### Protein complex isolation

Immunoprecipitation was done using GFP-trap magnetic agarose beads (chromotek, Planegg, Germany) according to manufacturer protocol.

Mass spectrometry – (this part of the experiment was performed through a collaboration with Prof. Tami Gieger's lab, Tel Aviv University).

### Sample preparation

Samples were washed 6 times with washing buffer containing 150 mM NaCl and 50 mM Tris-HCL (pH 7.5). In every wash, samples were gently rotated in an automatic rotator for 1 min. Then samples were collected using two rounds of 50 µl elution buffer. First samples were incubated in an elution buffer containing fresh 2 M urea, 50 mM Tris-HCL (pH7.5), and 1 mM DTT for 2 hr at room temperature and then collected in a new tube. Second, on the left beads, a second elution buffer containing 2 M urea, 50 mM Tris-HCL (pH7.5), and 5 mM Choloroacetmide was added for 10 min and collected the liquid to the same tube as collected with the first elution and wait 30 min. Then we added 5 µg/ml trypsin (Promega) for overnight incubation. Samples were collected in PCR tubes, vacuum dried, and re-suspended in 5 µl buffer containing 2% ACN and 0.1% trifluoroacetic acid (TFA).

### LC-MS

LC-MS/MS runs were performed on the EASY-nLC1000 UHPLC (Thermo Fisher Scientific) coupled to Q-Exactive HF mass spectrometers (Thermo Fisher Scientific). Peptides were separated with a 75 µM X 50 cm EASY-spray column (ThermoFisher Scientific) using a water-acetonitrile gradient of 120 min, with a flow rate of 300 ml/min at 40 °C. The resolutions of the MS and MS/MS spectra were 60,000 and 15,000, respectively. MS raw files of all samples were jointly analyzed by MaxQuant 2 version 1.6.2.6. MS/MS spectra were referenced to the Uniprot *Mus musculus* proteome.

### Statistical analysis

Analyses were performed using the Perseus software 3 version 1.6.2.3. For the student's t-test results we use a cut-off of FDR < 0.1, S0=0.1.

## RT-PCR

RAW 264.7 cells were transfected with either αCD19 scFv – GFP, GFP control, or additional control containing α chain of FcγRI receptor (CD64) attached to GFP. Transfections were done using JetOPTIMUS according to manufacturer's protocol (Polyplus Transfection). Twenty-four hours post-transfection, cells were collected from growth plate using trypsin (Promega). Cells lysis was performed and RNA was purified using RNA NucelSpin RNA manufacturer isolation protocol (Macherey-Nagel,

Dueren, Germany). To evaluate RNA sample quality, gel electrophoresis was performed to detect ribosomal RNA integrity. RNA concentration and purity were tested using NanoDrop One/One$^C$ Microvolume UV-Vis Spectrophotometer (Thermo Fisher Scientific) – all RNA samples used were at purity levels 260/280>2.0, 260/230 between 2.0 and 2.2.

cDNA was synthesized using qSript cDNA Synthesis Kit (Quantabio, Beverly, MA, USA).

| | | |
|---|---|---|
| GFP | Forward primer | AAGTTCAGCGTGTCCGGCGA |
| | Reverse primer | AAGCACTGCACGCCGTAGGT |
| Neomycin | Forward primer | TGAAGCGGGAAGGGACTGGC |
| | Reverse primer | CGAATGGGCAGGTAGCCGGA |
| *Mus musculus Actb* | Forward primer | GTCCACCTTCCAGCAGATGT |
| | Reverse primer | GCTCAGTAACAGTCCGCCT |

Quantitative Real-time PCR was conducted using PerfeCTa SYBR Green FastMix (Quantabio) using stepOne Real-Time PCR system. All experiments were done in triplicates.

## mRNA synthesis

For in vitro capped RNA synthesis pGEM4Z plasmid (kindly gifted from Prof. Eli Gilboa, University of Miami health system) containing T7 promoter along with insert αCD19 scFv – GFP or GFP for control was used as template.

For in vivo assay, we used βGlobin5'UTR (kindly gifted from Dr. Gal Cafri, Sheba medical center, Israel).

Synthesis was performed using AmpliCap-Max T7 High Yield Message Maker Kit (Cellscript, Maddison, WI, USA). The synthesis reaction product was tested for integrity via gel electrophoresis. mRNA was purified using NucelSpin RNA.

## mRNA transfection

A total of 5.0x10$^4$ RAW264.7 macrophages were plated on 24-well plate a day before the transfection. Synthesized mRNAs were transfected into RAW264.7 using jetMESSENGER RNA transfection kit (Polyplus Transfection) according to the manufacturer's protocol. Transfected cells were analyzed 16–24 hours' post-transfection.

For in vivo experiments, bone marrow cells were isolated from naïve 12 weeks CD45.1 female mice and incubated for seven days with GM-CSF and IL-4. 2.5x10$^6$ BMDC were collected in 200 μL OptiMEM and electroporated in 0.2 cuvettes using BioRad GenePulsar Xcell with 5 μg mRNA and 400 V square wave for one millisecond, as previously described (*Cafri et al., 2015*).

## T-cell proliferation assay

RAW 264.7 cells were transfected with OVA constructs using Lipofectamine 2000. Splenic T cells were isolated from OTI and OTII mice using Ficoll-paque plus density gradient (BD Biosciences, San Jose, CA, USA) followed by incubation with biotin-conjugated anti-CD8$^+$ and anti-CD4$^+$ magnetic beads and magnetic isolation using streptavidin-magnetic beads (BioLegend, Carlsbad, CA, USA). T cells were then labeled with CFSE, washed extensively three times in complete media, and co-cultured at ratios of 1:5 and 1:10 with transfected RAW264.7 cells. After three days, CSFE dilution in T cells was determined by CytoFLEX5.

## Confocal microscopy

For confocal microscopy imaging, cells were cultured on glass-bottom confocal plates (Cellvis, Mountain View, CA, USA). Cells were live-imaged or fixed and permeabilized with 2% PFA for 20 min. Fixed cultures were washed twice with PBS. For immunofluorescence, fixed cultures were blocked overnight with 5% BSA (ThermoFisher Scientific) and stained with 1:100 or 1:200 diluted primary antibodies. Cells were then washed with PBS containing 1% BSA and stained with a secondary antibody, diluted 1:100 or 1:200. Stains used included: Hoechst 33342 (ThermoFisher Scientific). ER staining kit - Cytopainter (Abcam). Anti-GPR78 BiP antibody (Abcam), G3BP1

**Table 1.** scFv construct design and preparation process.

| Construct subunits design | Construct preparation |
|---|---|
| (ssCD8α) – TA99 scFv – CH1 – hinge – CH2 – CH3 – transmembrane +intracellular portion of CD209 /C5aR−mCherry | All subunits were designed and designed and ordered in gBlock format from Integrated DNA Technologies (IDT) (Coralville, IA, USA). Subunits were added to pcDNA 3.1 (+) using Gibson assembly, and cloned to pmCherry- N1. |
| (ssCD8α) – TA99 scFv — CH3 – transmembrane +intracellular portion of CD209 /C5aR- mCherry | Construct was created by elimination of CH1-hinge-CH2 subunits through invert PCR followed by kinase, ligase, DpnI (KLD) enzyme mix. |
| pLVX: TA99 scFv – CH3 – (TM +IC) C5aR - mCherry | pLVX-IRES-Hyg vector was restricted using SpeI/NotI. Receptor insert including TA99 scFv - heavy chain - C5aR (transmembrane +intracellular) - mCherry was restricted from mCherry vector via restriction enzymes NheI/NotI followed by ligation using 4T DNA ligase. |
| (ssCD8α) – CH1 – hinge – CH2 – CH3 – transmembrane +intracellular portion of CD209 /C5aR- mCherry | Constant heavy sequence was isolated and amplified using invert PCR for scFv removal followed by KLD enzymes. |
| (ssCD8α) – TA99 scFv - transmembrane and intracellular portion of either CD209 or C5aR- mCherry | Removal of constant heavy subunits was done by invert PCR amplification. Followed by KLD enzyme reaction. |
| (ssCD8α) – TA99 scFv – CD8α hinge – CD8α transmembrane – mCherry | Extracellular portion was isolated via restriction with BamHI/EcoRI and inserted into pmCherryN1 vector followed by ligation with 4T DNA ligase. |
| (ssCD8α) –CD8α hinge - a transmembrane and intracellular portion of either CD209 or C5aR - mCherry | scFv portion was removed via invert PCR using Fw primer that included a tail of CD8α hinge sequence as an extracellular remnant. The PCR product was used in KLD enzymes mix reaction. |
| TA99 scFv - GFP | TA99 scFv sequence was restricted from pemCherry-N1 plasmid using EcoRI/XhoI enzymes and inserted to peGFP-N1 plasmid linearized with same enzymes. |
| GFP - T2A - TA99 scFv - mCherry | T2A sequence was designed and ordered in linear gBlock sequence, flanked by restriction enzymes: BglII/ NheI. The sequence was inserted to peGFP-C1 vector. GFP-T2A sequence was isolated via restriction with NheI/XhoI and inserted to plasmid which included an insert of TA99 scFv – mCherry. |
| GFP – TA99 scFv – mCherry | Removal of T2A sequence was performed by invert PCR followed by KLD enzyme mix reaction. |
| (ssCD8α) – αCD19 scFv – CD8α hinge – CD8α transmembrane – mCherry | (ssCD8α) – TA99 scFv – CD8α hinge – CD8α transmembrane – mCherry was used as backbone. TA99was removed by invert PCR amplification exclusion ofαCD19 scFv was amplified using plasmid pHR-PGK-antiCD19-synNotch-GalVP64 (plasmid#79125) which was purchased from Addgene (Watertown, MA, USA). Both backbone and insert were flanked with XhoI/EcoRI. |
| αCD19 scFv - GFP | αCD19 scFv from plasmid#79125 (Addgene) was amplified using primers with tails encoding restriction sites for XhoI/EcoRI enzymes. peGFP-N1 backbone was restricted using same enzymes. |
| pGEM4Z: T7 promoter - αCD19 scFv – GFP – polyA tail | αCD19 scFv was cloned into pGEM4Z using restriction enzymes NheI +NotI. pGEM4Z GFP was restricted using: XbaI +NheI. |
| pLVX: αCD19 scFv - GFP | αCD19 – GFP was isolated from peGFP-N1 backbone using enzymes NheI/NotI and inserted into pLVX backbone linearized using SpeI/NotI. |
| GFP - αCD19 scFv | αCD19 scFv sequence was isolated and amplified using primers which included overhang sequence of restriction sites for enzymes XhoI/EcoRI. peGFPC1 was restricted with same enzymes followed by ligation. |
| TA99 variable heavy chain-GFP | Variable heavy chain was isolated and amplified using invert PCR for variable light chin removal followed by KLD enzyme mix. |
| TA99 variable light chain-GFP | Variable light chain was isolated and amplified using invert PCR for variable heavy chin removal followed by KLD enzyme mix. |
| αCD19 variable heavy chain-GFP | Variable heavy chain was isolated and amplified using invert PCR for variable light chin removal followed by KLD enzyme mix. |
| αCD19 variable light chain-GFP | Variable light chain was isolated and amplified using invert PCR for variable heavy chin removal followed by KLD enzyme mix. |
| αCD19 mutated VL (linear) | G-block of mutated αCD19 variable light chain. Two-point mutations included replacement of Cysteine amino acid with glycine in positions 24 and 89, flanked by restriction enzymes XhoI/EcoRI was designed and orders from IDT. PeGFP-N1 plasmid was restricted with same enzymes. |
| αCD19 mutated VH (linear) | gBlock of mutated αCD19 variable heavy chain. Two-point mutations included replacement of Cysteine amino acid with glycine in positions 22 and 95, flanked by restriction enzymes XhoI/EcoRI was designed and orders from IDT. PeGFP-N1 plasmid was restricted with same enzymes. |

*Table 1 continued on next page*

*Table 1 continued*

| Construct subunits design | Construct preparation |
|---|---|
| Immunoglobulin loop fragment GFP | DNA sequence was ordered in linear gBlock from IDT. Insert of interest was flanked by XhoI/EcoRI and inserted into peGFP-N1 backbone linearized with both enzymes. Final amino acid sequence included: MVTISCRASQDISKYL-GGGGSGGGGSGGGGS-EQEDIATYFCQQGN |
| (ssCD8α) – CD64 – transmembrane +intracellular portion of C5aR/CD209/μ – GFP | CD64 sequence was purchased in gBlock format from IDT, subunits were added in sequence keeping reading frame in check using homology sequences into peGFP-N1. |
| (ssCD8α) – CD64 – transmembrane +intracellular portion of C5aR/CD209/μ – T2A - GFP | T2A sequence was ordered in linearized DNA gBlock from IDT. Sequence was flanked by homology sequence which overlapped linearized plasmid ends created by restriction with enzymes BamHI/XmaI. |
| pLVX: (ssCD8α) – CD64 – transmembrane +intracellular portion of C5aR /CD209 – T2A GFP | Receptor sequence including all subunits - GFP was isolated from peGFPN1 plasmid using restriction enzymes NheI/NotI. pLVX backbone was linearized using SpeI/NotI. |
| CD64 (extracellular +transmembrane) – GFP (used for RT PCR expression control) | Transmembrane +intracellular sequences were removed from (ssCD8α) – CD64 – transmembrane +intracellular sequence – GFP using invert PCR. Followed by KLD enzyme's reaction. |
| Alpha mu in βGlobin5'UTR | Plasmid was ordered with insert in place through A2S technologies Ltd (Yavne, Israel) |
| anti TRP1 murine TCR | Minigene was cloned to pcDNA3-EGFP backbone plasmid using restriction enzymes. |
| TA99 scFv – OVA$_{257-264}$ – G4S linker - OVA$_{323-339}$ | DNA sequence including all subunits was designed and ordered in gBlock via IDT. Sequence was inserted into pcDNA 3.1 (+) using restriction enzymes. |
| OVA$_{257-264}$ – G4S linker - OVA$_{323-339}$ | Control construct to scFv-OVA was prepared by removal of scFv sequence using invert PCR amplification followed by a KLD reaction according to protocol. |

antibody (mab#17798, Cell signaling technologies), and anti-mouse CD64 (FcγRI) (Biolegend). DAPI (Abcam). All specimens were imaged by ZEISS LSM800 (ZEISS) and analysed by ZEN 2.3 (ZEISS) and ImageJ.

## Structured illumination microscopy

μ–Slide 8 well High Grid 500 (ibidi - Munich, Germany) were pre-coated with trastuzumab (Roche) and left to incubate overnight at 4 °C then washed thrice with PBS. transfected RAW 264.7 cells were cocultured with 4T1 cell line expressing human HER2. Immunological synapse was visualized using 3i Marianas Spinning Disk Confocal (Denver, CO, USA). For Granzyme B detection, cells were fixated using PFA 4% and washed to allow permeabilization and stained using Alexa Fluor 647 anti-human/mouse Granzyme B Antibody (Biolegend 515405).

## Flow cytometry

Flow cytometry was performed on CytoFLEX5 (Beckman Coulter, Brea, CA, USA) using anti-mouse CD64 (FcγRI) (clone: X54-5/7.1), anti-mouse CD3 (clone: 17-A2) anti- mouse CD45 (clone: 30-F11), Annexin V, and propidium iodide (PI) (all purchased from Biolegend). Data sets were analyzed using FlowJo software (Tree Star, Inc).

## Killing assay

For measuring tumor cell killing, $10^4$ 4T1 H2B-tdTomato, which are genetically modified to express human EGFR or HER2 on cancer cell membranes, were cultured in 96 well plates for 2 hr to adhere. Transfected myeloid cells were added at E:T ratios of 4:1 and 8:1, with or without 10 μg/mL of Trastuzumab (Roche, Basel, Switzerland), Cetuximab (Merck, Darmstadt, Germany), and Rituximab (Genentech, San Francisco, CA, USA). Cells were imaged by IncuCyte S3 imager (Sartorius, Göttingen, Germany), and the numbers of target cells were calculated by incuCyte software. In some experiments, cells were stained with Annexin V and propidium iodide (PI) (Biolegend) according to manufacturer's instructions, and specific tumor cell lysis was measured by CytoFLEX5.

## Constructs subunits and design process

Detailed description of constructs design and subunits, as well as preperation process are demonstrated in *Table 1*.

## Kits and materials used for cloning

NucleoSpin Plasmid easy pure (Macherey-Nagel), NucleoSpin Gel and PCR clean-up (Macherey-Nagel), NucleoBond Xtra Maxi (Macherey-Nagel), Zymoclean Gel DNA recovery kit (Zymo Research, Irvine, CA, USA). *Cloning*: DH5α Competent cells (ThermoFisher scientific) were grown in LB broth with agar (Miller) (Sigma Aldrich) or LB broth (Miller) (Merck) supplemented with Ampicillin, sodium salt or Kanamycin sulfate Bio Basic, Markham, Ontario, Canada. *Enzymes and reagents*: Restriction enzymes, T4 DNA ligase, KLD enzyme mix, and Phusion High Fidelity DNA polymerase were purchased from New England biolabs (Ipswich, MA, USA), Taq DNA polymerase mix (PCRBIO, Wayne, PA, USA), RepliQa HiFi Assembly Mix (Quantabio). *Gel electrophoresis*: SeaKem LE agarose (Lonza, Basel, Switzerland), Ethidium Bromide (Merck), TAE buffer (Sartorius, Göttingen, Germany).

## Statistical analyses

All statistical analyses and graphs were performed by GraphPad Prism 9. Each experimental group consisted of at least three mice and repeated at least three independent times. Significance of results was determined using the nonparametric one-way ANOVA with Tukey's and Sidak correction, when multiple groups are analyzed, two-way ANOVA with Tukey's and Sidak correction for multiple comparisons.

## Additional information

### Competing interests

Diana Rasouluniriana: This paper was funded by Gilboa Therapeutics where Diana Rasouluniriana is a shareholder. Peleg Rider: This paper was funded by Gilboa Therapeutics where Peleg Rider is a shareholder. Yaron Carmi: This paper was funded by Gilboa Therapeutics where Yaron Carmi is a shareholder. The other authors declare that no competing interests exist.

### Funding

| Funder | Grant reference number | Author |
| --- | --- | --- |
| Fritz Thyssen Stiftung | | Yaron Carmi |
| Israel Cancer Research Fund | | Yaron Carmi |
| Israel Science Foundation | | Yaron Carmi |

The funders had no role in study design, data collection and interpretation, or the decision to submit the work for publication.

### Author contributions

Leen Farhat-Younis, Conceptualization, Investigation, Methodology, Writing - original draft, Project administration, Writing - review and editing; Manho Na, Investigation, Methodology; Amichai Zarfin, Investigation; Aseel Khateeb, Alon Richter, Annette Gleiberman, Methodology; Nadine Santana-Magal, Conceptualization, Validation; Amit Gutwillig, Conceptualization, Methodology; Diana Rasouluniriana, Investigation, Visualization; Lir Beck, Formal analysis, Visualization; Tamar Giger, Data curation, Formal analysis; Avraham Ashkenazi, Conceptualization; Adi Barzel, Conceptualization, Investigation; Peleg Rider, Conceptualization, Investigation, Methodology; Yaron Carmi, Conceptualization, Supervision, Investigation, Methodology, Writing - original draft

### Author ORCIDs

Leen Farhat-Younis ⓘ http://orcid.org/0000-0001-8622-6788
Yaron Carmi ⓘ http://orcid.org/0000-0002-0972-0616

### Ethics

This study was performed in strict accordance with the recommendations in the Guide for the Care and Use of Laboratory Animals of the National Institutes of Health. All of the animals were handled according to approved institutional animal care and use committee protocols of Tel Aviv University. The protocol was approved by the Committee on the Ethics of Animal Experiments of Tel Aviv University (Permit Number: TAU - MD - IL - 2310 - 163 - 5 and 01-17-067). Every effort was made to minimize suffering.

Reviewer #2 (Public Review): https://doi.org/10.7554/eLife.91999.3.sa1
Author response https://doi.org/10.7554/eLife.91999.3.sa2

## Additional files

### Supplementary files

• MDAR checklist

### Data availability

Source data files have been provided for *Figures 1, 4 and 5*.

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

# Appendix 1

## Appendix 1—key resources table

| Reagent type (species) or resource | Designation | Source or reference | Identifiers | Additional information |
|---|---|---|---|---|
| Strain, strain background (57BL/6) | Wild-type (WT) C57BL/6 mice | Envigo | Strain #:000664 | |
| Strain, strain background (129S1) | 129S1/SvlmJ | The Jackson Laboratory | Strain #:002448 | |
| Strain, strain background (OT-I) | C57BL/6-Tg(TcraTcrb)1100Mjb/J OT-I | The Jackson Laboratory | Strain #:003831 | |
| Strain, strain background (CD45.1) | B6.SJL-*Ptprc$^a$ Pepc$^b$*/BoyJ | The Jackson Laboratory | Strain #:002014 | |
| Strain, strain background (OT-II) | B6.Cg-Tg(TcraTcrb)425Cbn/J OT-II | The Jackson Laboratory | Strain #:004194 | |
| Cell line (Homo-sapiens) | Human Embryonic Kidney (HEK)–293 FT cells | ThermoFisher Scientific | R70007 | |
| Cell line (*Mus musculus*) | DC 2.4 | Merck Milipore | SCC142 | |
| Cell line (*Mus musculus*) | RAW 264.7 | ATCC | TIB-71 | |
| Cell line (Homo-sapiens) | Jurkat | ATCC | TIB-152 | |
| Cell line (Homo-sapiens) | THP-1 | ATCC | TIB-202 | |
| Cell line (*Mus musculus*) | B16F10 | ATCC | CRL-6475 | |
| Cell line (*Mus musculus*) | 4T1 | ATCC | CRL-2539 | |
| Transfected construct (4T1, *Mus musculus*) | 4T1 coexpressing H2b-Tdt / HER2 | This paper | | Cell line was created using lentiviral vectors |
| Antibody | anti-mouse CD64 (FcγRI) (mouse monoclonal) | Biolegend | Cat# 139315 clone: X54-5/7.1 | 1 µl in 100 µl volume |
| Antibody | anti-mouse CD3 (rat monoclonal) | Biolegend | Cat# 100201 Clone: 17-A2 | 0.5 µl in 100 µl volume |
| Antibody | anti-mouse CD45 (rat monoclonal) | Biolegend | Cat# 103101 Clone: 30-F11 | 0.5 µl in 100 µl volume |
| Antibody | Fluorochrome-conjugated antibodies against phospho-p38 (Thr180/Tyr182) (rabbit monoclonal) | Cell Signaling Technologies | Cat# 8623 | Dilution 1:50 |
| Antibody | Fluorochrome-conjugated antibodies against phospho-JNK (Ser63) (rabbit monoclonal) | Cell Signaling Technologies | Cat# 91952 | 1:100 |
| Antibody | phospho-ERK1/2 (p44) (pT202/pY204) (rabbit monoclonal) | Cell Signaling Technologies | Cat# 14095 | 1:50 |
| Antibody | phospho-Akt (pY473) (rabbit polyclonal) | Cell Signaling Technologies | Cat# 9271 | 1:100 |
| Recombinant DNA reagent | (ssCD8α) – TA99 scFv – CH1 – hinge – CH2 – CH3 – transmembrane +intracellular portion of CD209 /C5aR-mCherry | This paper | | This construct was prepared in lab using linear DNA ordered from Integrated DNA Technologies (IDT). Detailed construct preparation can be found in *Table 1* in main article file. |

*Appendix 1 Continued on next page*

*Appendix 1 Continued*

| Reagent type (species) or resource | Designation | Source or reference | Identifiers | Additional information |
|---|---|---|---|---|
| Recombinant DNA reagent | (ssCD8α) – TA99 scFv — CH3 – transmembrane +intracellular portion of CD209 /C5aR- mCherry | This paper | | This construct was prepared in lab. Detailed construct preparation can be found in *Table 1* in main article file. |
| Recombinant DNA reagent | pLVX: TA99 scFv – CH3 – (TM +IC) C5aR - mCherry | This paper | | This construct was prepared in lab. Detailed construct preparation can be found in *Table 1* in main article file. |
| Recombinant DNA reagent | (ssCD8α) – CH1 – hinge – CH2 – CH3 – transmembrane +intracellular portion of CD209 /C5aR- mCherry | This paper | | This construct was prepared in lab. Detailed construct preparation can be found in *Table 1* in main article file. |
| Recombinant DNA reagent | (ssCD8α) – TA99 scFv - transmembrane and intracellular portion of either CD209 or C5aR- mCherry | This paper | | This construct was prepared in lab. Detailed construct preparation can be found in *Table 1* in main article file. |
| Recombinant DNA reagent | (ssCD8α) – TA99 scFv – CD8α hinge – CD8α transmembrane – mCherry | This paper | | This construct was prepared in lab. Detailed construct preparation can be found in *Table 1* in main article file. |
| Recombinant DNA reagent | (ssCD8α) –CD8α hinge - a transmembrane and intracellular portion of either CD209 or C5aR - mCherry | This paper | | This construct was prepared in lab. Detailed construct preparation can be found in *Table 1* in main article file. |
| Recombinant DNA reagent | TA99 scFv - GFP | This paper | | This construct was prepared in lab. Detailed construct preparation can be found in *Table 1* in main article file. |
| Recombinant DNA reagent | GFP - T2A - TA99 scFv - mCherry | This paper | | This construct was prepared in lab using linear DNA ordered from (IDT). Detailed construct preparation can be found in *Table 1* in main article file. |
| Recombinant DNA reagent | GFP – TA99 scFv – mCherry | This paper | | This construct was prepared in lab. Detailed construct preparation can be found in *Table 1* in main article file. |
| Recombinant DNA reagent | (ssCD8α) – αCD19 scFv – CD8α hinge – CD8α transmembrane – mCherry | This paper | | This construct was prepared in lab. Detailed construct preparation can be found in *Table 1* in main article file. |
| Recombinant DNA reagent | αCD19 scFv - GFP | This paper | | This construct was prepared in lab. Detailed construct preparation can be found in *Table 1* in main article file. |
| Recombinant DNA reagent | pGEM4Z: T7 promoter - αCD19 scFv – GFP – polyA tail | This paper | | This construct was prepared in lab. Detailed construct preparation can be found in *Table 1* in main article file. |
| Recombinant DNA reagent | pLVX: αCD19 scFv - GFP | This paper | | This construct was prepared in lab. Detailed construct preparation can be found in *Table 1* in main article file. |
| Recombinant DNA reagent | GFP - αCD19 scFv | This paper | | This construct was prepared in lab. Detailed construct preparation can be found in *Table 1* in main article file. |
| Recombinant DNA reagent | TA99 variable heavy chain-GFP | This paper | | This construct was prepared in lab. Detailed construct preparation can be found in *Table 1* in main article file. |
| Recombinant DNA reagent | TA99 variable light chain-GFP | This paper | | This construct was prepared in lab. Detailed construct preparation can be found in *Table 1* in main article file. |
| Recombinant DNA reagent | αCD19 variable heavy chain-GFP | This paper | | This construct was prepared in lab. Detailed construct preparation can be found in *Table 1* in main article file. |

*Appendix 1 Continued on next page*

*Appendix 1 Continued*

| Reagent type (species) or resource | Designation | Source or reference | Identifiers | Additional information |
|---|---|---|---|---|
| Recombinant DNA reagent | αCD19 variable light chain-GFP | This paper | | This construct was prepared in lab. Detailed construct preparation can be found in *Table 1* in main article file. |
| Recombinant DNA reagent | αCD19 mutated VL (linear) | This paper | | This construct was prepared in lab using linear DNA ordered from (IDT). Detailed construct preparation can be found in *Table 1* in main article file. |
| Recombinant DNA reagent | αCD19 mutated VH (linear) | This paper | | This construct was prepared in lab using linear DNA ordered from (IDT). Detailed construct preparation can be found in *Table 1* in main article file. |
| Recombinant DNA reagent | Immunoglobulin loop fragment GFP | This paper | | This construct was prepared in lab using linear DNA ordered from (IDT). Detailed construct preparation can be found in *Table 1* in main article file. |
| Recombinant DNA reagent | (ssCD8α) – CD64 – transmembrane +intracellular portion of C5aR/CD209/μ – GFP | This paper | | This construct was prepared in lab using linear DNA ordered from (IDT). Detailed construct preparation can be found in *Table 1* in main article file. |
| Recombinant DNA reagent | (ssCD8α) – CD64 – transmembrane +intracellular portion of C5aR/CD209/μ – T2A - GFP | This paper | | This construct was prepared in lab using linear DNA ordered from (IDT). Detailed construct preparation can be found in *Table 1* in main article file. |
| Recombinant DNA reagent | pLVX: (ssCD8α) – CD64 – transmembrane +intracellular portion of C5aR /CD209 – T2A GFP | This paper | | This construct was prepared in lab. Detailed construct preparation can be found in *Table 1* in main article file. |
| Recombinant DNA reagent | CD64 (extracellular +transmembrane) – GFP | This paper | | This construct was prepared in lab. Detailed construct preparation can be found in *Table 1* in main article file. |
| Recombinant DNA reagent | Alpha mu in βGlobin5'UTR | This paper | | This construct was designed in lab and ordered in plasmid from A2S technologies |
| Recombinant DNA reagent | anti TRP1 murine TCR | This paper | | This construct was prepared in lab. Detailed construct preparation can be found in *Table 1* in main article file. |
| Recombinant DNA reagent | anti TRP1 murine TCR | This paper | | This construct was prepared in lab. Detailed construct preparation can be found in *Table 1* in main article file. |
| Recombinant DNA reagent | TA99 scFv – OVA$_{257-264}$ – G4S linker - OVA$_{323-339}$ | This paper | | This construct was prepared in lab using linear DNA ordered from (IDT). Detailed construct preparation can be found in *Table 1* in main article file. |
| Recombinant DNA reagent | OVA$_{257-264}$ – G4S linker - OVA$_{323-339}$ | This paper | | This construct was prepared in lab. Detailed construct preparation can be found in *Table 1* in main article file. |
| Commercial assay, kit | Mouse IgG ELISA Kit | Bethyl | Catalog # E99-131 | |
| Commercial assay, kit | Mouse IgM ELISA Kit | Bethyl | Catalog # E99-101 | |
| Commercial assay, kit | Human Granzyme B ELISA Kit | Abcam | ab235635 | |
| Commercial assay, kit | Griess Reagent System | Promega | Catalog: G2930 | |
| Chemical compound, drug | Trastuzumab | Roche | INF/INJ-HER-2021 02–0 | |

*Appendix 1 Continued on next page*

*Appendix 1 Continued*

| Reagent type (species) or resource | Designation | Source or reference | Identifiers | Additional information |
|---|---|---|---|---|
| Chemical compound, drug | Cetuximab | Merck | Erbitux 5 mg/mL solution for infusion | |
| Chemical compound, drug | Rituximab | Genentech | Rituxan | |
| Other | FITC Annexin V | Biolgend | Cat: 640905 | 5 µl in 100 µl volume |
| Other | Propidium iodide (PI) | Biolegend | Cat: 421301 | 0.5 mg/ml |

