## [Editor Report · eLife assessment]

The findings are **fundamental** for understanding IgM signaling in myeloid cells. The work is **compelling** in its ability to manipulate and harness myeloid cells to further anti-tumor immunity.

---

## [Referee Report · Reviewer #2 (Public Review)]

Summary:

While a significant portion of immunotherapy research has focused on the pivotal role of T cells in tumor immunity, their effectiveness may be limited by the suppressive nature of the tumor environment. On the other hand, myeloid cells are commonly found within tumors and can withstand these adverse conditions. However, these cells often adopt an immunosuppressive phenotype when infiltrating tumors. Therefore, manipulating myeloid cells could potentially enhance the anti-tumor potential of immunotherapy.

In this manuscript, Farhat-Younes and colleagues have demonstrated that activating the IgM receptor signaling in myeloid cells induces an oxygen burst, the secretion of Granzyme B, and the lysis of adjacent tumor cells. Furthermore, they have outlined a strategy to utilize these features to generate CAR macrophages. However, they have identified a limitation: the expression of scFv in myeloid cells induces ER stress and the degradation of misfolded proteins. To address this issue, chimeric receptors were designed based on the high-affinity FcγRI for IgG. When macrophages transfected with these receptors were exposed to tumor-binding IgG, extensive tumor cell killing, and the release of reactive oxygen species and Granzyme B were observed.

Strengths:

In general, I consider this work to be significant, and the results are compelling. It emphasizes the specific considerations and requirements for successful manipulation in myeloid cells, which could further advance the field of cellular engineering for the benefit of immunotherapy

Following the revision of the original manuscript, I can clearly state that my concerns have been addressed and the article has been improved.

---

## [Author Response]

The following is the authors’ response to the original reviews.

**Reviewer #1 (Public Review):**
Farhat-Younis and colleagues demonstrate tumor-specific IgM's capacity to induce tumor cell death in monocyte-derived dendritic cell cultures. They subsequently designed a chimeric receptor based on high-affinity FcgRI. However, the authors found that the transfection process was more efficient when either the variable light or heavy chain was transfected individually rather than the entire scFv. This scFv construct led to an endoplasmic reticulum (ER) stress response and scFv degradation. A considerable portion of the manuscript is dedicated to the negative scFv expression results. The authors pivoted to a modified FcgRI capable of transmitting IgM signals. This represents a tremendous amount of work in the development of this chimeric receptor, the critical experiment showing efficacy in vivo was not presented, and instead various in vitro assays are shown. Thus, this manuscript will markedly benefit from showing improved responses to tumors in vivo when macrophages express FcgRI-IgM.

We deeply thank the reviewer for his thoughtful comments and overall favorable review of our manuscript.

1. In a mouse tumor model, the authors demonstrated that monocyte-derived dendritic cells (MoDCs) treated with IgG immune complexes (ICs) were more effective at preventing tumor growth compared to those treated with IgM ICs (as shown in Figure 1B). In Figure 1C, their in vitro experiments revealed that IgM resulted in tumor cell death, as well as increased production of nitric oxide (NO) and granzyme B. How do the authors reconcile IgG IC-treated MoDCs performing better in preventing tumors in vivo than IgM IC-treated MoDCs, despite the in vitro results with IgM-ICs. The authors speculate that IgG IC-treated MoDCs might trigger T cell immunity but do not show T cell involvement.

We apologize for not making this point clearer. We have extensively studied this phenomenon and published two papers that detailed the underlying mechanism in two consecutive papers (PMID: 27812544, PMID: 25924063). Briefly, we showed that DC activated with IgM-IC DC undergo cell death concomitantly to their release of lytic granules and lysis of tumor cells. As a result, they do not migrate to the lymph nodes where they should induce reactive T cell clones. In contrast, DC activated with IgG-IC do not elicit in vitro cytotoxicity but rather process the IC to present its derived antigens of MHC-II. We addressed that issue in the revised version and cited the relevant paper to further clarify it.

(2) The authors report distinct functional consequences of MoDCs incubated with tumor-IgG complexes and tumor IgM complexes. Tumor growth was inhibited and T cell immunity induced with the former. The latter, however, elicited robust anti-tumor killing. What happens if MoDCs are incubated with both IgG and IgM complexes? If this combined treatment induces effective killing and T cell memory, would this impact the design of the chimeric receptor to include IgG responsiveness as well?

This is a very interesting point. As mentioned above, our previous publications strongly suggest that tumor binding IgG and IgM induce different processes in myeloid cells. Yet, since MoDC naturally express the high affinity receptors for IgG FcgRI, we speculate that treating tumor-bearing mice modified monocyte, alone or in combination with tumor-binding IgG, would shed some light into that. Indeed, such treatment elicit a strong T cell immunity in these mice and the data was added to Supplementary Data Figure S4J. With that being said, a complete analysis of this question is very complicated and extent beyond the scope of this work. We would like to emphasize that the purpose of this work is to highlight some of the challenges unique to genetic manipulation in myeloid cells and to suggest one alternative scaffold for integrating signaling in these cells. We do not argue that the specific solution presented here is the most potent one and more work is required before promoting such treatment into the clinic. We have added a sentence to the Discussion section that stress that issue.

(3) In Figure 5H, the authors demonstrate the ability of the chimeric receptor construct to deplete tumor cells in vitro. The ms would improve if the authors could show the chimeric receptor construct results in tumor cell death and/or prevention in an in vivo model. Similarly, if combined stimulation with IgG and IgM complexes enhances tumor response, this should be incorporated into the therapeutic strategy.

This is a wonderful suggestion. To address that, we challenged C57Bl/6 mice with B16F10 melanoma and allowed them to grow until it reached a palpable size of approximately 25 mm2. Concomitantly, we cultured bone marrow dendritic cells from syngeneic mice and transfected them with a linear mRNA of the alpha/mu construct. Tumor bearing mice were then treated with alpha/mu and sham transduced BMDC alone, or in combination with antibody against the melanoma antigen Trp1 (TA99). The results were added as Figure 5K and to Supplementary Figure S4h-S4I.

**Reviewer #2 (Public Review):**
Summary:While a significant portion of immunotherapy research has focused on the pivotal role of T cells in tumor immunity, their effectiveness may be limited by the suppressive nature of the tumor environment. On the other hand, myeloid cells are commonly found within tumors and can withstand these adverse conditions. However, these cells often adopt an immunosuppressive phenotype when infiltrating tumors. Therefore, manipulating myeloid cells could potentially enhance the anti-tumor potential of immunotherapy.In this manuscript, Farhat-Younes and colleagues have demonstrated that activating the IgM receptor signaling in myeloid cells induces an oxygen burst, the secretion of Granzyme B, and the lysis of adjacent tumor cells. Furthermore, they have outlined a strategy to utilize these features to generate CAR macrophages. However, they have identified a limitation: the expression of scFv in myeloid cells induces ER stress and the degradation of misfolded proteins. To address this issue, chimeric receptors were designed based on the high-affinity FcγRI for IgG. When macrophages transfected with these receptors were exposed to tumor-binding IgG, extensive tumor cell killing, and the release of reactive oxygen species and Granzyme B were observed.Strengths:In general, I consider this work to be significant, and the results are compelling. It emphasizes the specific considerations and requirements for successful manipulation in myeloid cells, which could further advance the field of cellular engineering for the benefit of immunotherapy

We thank the reviewer for his thoughtful comments and overall appreciation of our findings.

Weaknesses:Nevertheless, there are several minor issues that should be addressed:(1) TCR fragments are commonly used to induce ER stress in non-immune cells. Therefore, it would be interesting to investigate whether TCR fragments can be expressed in myeloid cells and if they induce ER stress. Addressing this issue would support the notion that these cells lack the ER chaperones required for folding immunoglobulin variable chains.

This is a wonderful suggestion. To assess that possibility, we cloned the alpha chain of anti-Trp1 TCR and transfected RAW 264.7 macrophages. Importantly, we could not detect expression on this construct in macrophages, further supporting our findings with scFv in these cells. We added this result to Figure 4J and Supplementary Figure S3C.

(2) It would be valuable to determine whether, after the degradation of scFv fragments by myeloid cells, they are presented on MHC-I and MHC-II.

This is a very interesting point. To address that, we generated a genetic construct where we fused the anti-CD19 scFv to a polypeptide composed from the MHCI and the MHCII fragments of Ova Albumin. Next, DC 2.4 were transfected with this construct and measured their capacity to stimulate the proliferation of CD8+ T cells from OT-I and CD4+ from OT-II mice. DC transfected with this construct efficiently stimulated the proliferation of both T cells, suggesting that both Ova fragments are indeed presented on MHCI and MHCII. Nonehteless, DC transfected with polypeptide of MHCI and MHCII fragments of Ova Albumin only (with no scFv), were almost equally effective in stimulating OT-I and OT-II T cell proliferation. We added that result to Supplementary Figure S3D-S3E.

(3) Some methodological details, such as the vaccination protocol and high-resolution microscopy procedures, are missing from the text.

We thank the reviewer for pointing out these issues. We added the missing details to the revised version of the manuscript.